

# Socioeconomic status and risk of gestational diabetes mellitus among pregnant women in Zunyi, China

Yan Xie[1,*], Jing Jiang[1,*], Jing Yang[1], Shiyi Gao[1], Rong Zeng[1], Yijun Liu[1], LingLu Wang[2], Pei Xu[1], Kunming Tian[1], Shimin Xiong[1], Xingyan Liu[1], Xubo Shen[1], Hong Pan[1] and YuanZhong Zhou[1]

[1] Zunyi Medical University, Zunyi, China
[2] The Second Affiliated Hospital of Zunyi Medical University, Zunyi, China
[*] These authors contributed equally to this work.

## ABSTRACT

**Background**. The impact of socioeconomic status on disease is becoming increasingly apparent. However, the relationship between gestational diabetes mellitus (GDM) and socioeconomic status (SES) has been less studied and remains inconsistent. The aim of this study was to investigate the relationship between SES and GDM.

**Methods**. All participants were selected from a prospective study on maternal and infant health in Zunyi City, China, between 2020 and 2022. Data on educational attainment, occupation, and household income were collected through standard questionnaires administered during face-to-face interviews. Logistic regression models were used to calculate adjusted odds ratios (ORs) and 95% confidence intervals (CIs), evaluating the association between GDM and SES—a composite measure comprising educational attainment, occupational status, and household monthly income.

**Results**. Among 5,868 participants, 690 women (11.8%) were diagnosed with GDM. After adjusting for potential confounders, no significant association was observed between educational attainment and GDM prevalence. Compared to public sector employees, women engaged in private sector occupations, freelancing, or farming exhibited a lower GDM risk. Household monthly income demonstrated a significant positive correlation with GDM incidence. Stratified analyses revealed distinct age-related patterns: higher education attainment was associated with reduced GDM risk in women aged >35 years, while occupational influences on GDM were more pronounced in this age group. Conversely, income effects were stronger among women aged ≤35 years. BMI stratification further indicated that occupational factors predominantly affected GDM risk in underweight women (BMI <18.5 kg/m$^2$), whereas income exhibited stronger associations in women with BMI ≥18.5 kg/m$^2$.

**Conclusion**. Our study revealed a significant association SES and GDM development. Among household income and occupation emerged as stronger predictor of GDM to educational attainment in Zunyi City, Guizhou province, China.

Corresponding authors
Hong Pan, ph-tian999@163.com
YuanZhong Zhou,
zhouyuanzhong@zmu.edu.cn

## INTRODUCTION

Gestational diabetes mellitus (GDM) defined as the first diagnosis of impaired glucose tolerance during pregnancy during pregnancy, represents one of the most prevalent pregnancy-related metabolic disorders (*Zhou et al., 2021*). Rapid economic development in China has been paralleled by lifestyle shifts toward high-sugar and high-fat dietary patterns, contributing to a dramatic surge in GDM prevalence from 4% in 2010 to 21% in 2020 (*Zhu et al., 2022*). Although gestational glucose intolerance typically resolves postpartum, GDM confers substantial clinical burdens: affected women exhibit elevated risks of pregnancy complications (*e.g.*, preeclampsia) and long-term metabolic sequelae including diabetes and cardiovascular diseases (*Harreiter, Dovjak & Kautzky-Willer, 2014*; *You et al., 2021*; *Yogev, Xenakis & Langer, 2004*). Furthermore, offspring of mothers with GDM demonstrate increased susceptibility to childhood obesity, metabolic syndrome, and early-onset type 2 diabetes mellitus (T2DM) (*Hillier et al., 2007*; *Dabelea et al., 2000*; *Clausen et al., 2009*). So early identification of risk factors for GDM has important public health and clinical implications, helping to prioritize interventions to reduce adverse health outcomes for mothers and their offspring.

Current research on GDM predominantly focuses on diagnostic criteria, risk stratification, and clinical management. However, the association between socioeconomic status (SES)—a composite measure of education, occupation, and household income reflecting an individual's societal position—and GDM risk remains underexplored in specific regional contexts, particularly in developing areas. While existing studies from high-income regions (*e.g.*, Finland *Roustaei et al., 2023*) and urban Chinese populations (*e.g.*, Tianjin *Handelsman et al., 2015*) suggest low SES correlates with elevated GDM risk, these findings primarily attribute the association to pre-pregnancy body mass index (BMI) and focus on singular SES indicators. Notably, evidence from economically disadvantaged regions remains scarce, with limited investigation into how multidimensional SES components collectively influence GDM epidemiology.

As a socioeconomically transitional region in Southwest China, Zunyi City presents a unique case study. Its distinct geographic characteristics and evolving economic landscape necessitate localized research to elucidate SES-GDM dynamics. Investigating this relationship not only informs region-specific prevention policies but also generates transferable insights for similar developing regions. This study therefore aims to: (1) quantify the association between multidimensional SES indicators (educational attainment, occupational status, and household income) and GDM risk in Zunyi; (2) identify critical intervention points through stratified analysis of maternal health outcomes across socioeconomic subgroups; and (3) propose evidence-based strategies to mitigate disparities in maternal-infant health outcomes.

## METHODS

### Study participants

This study utilized data from an ongoing prospective Maternal-Infant Health Cohort Study conducted in Zunyi City, Guizhou province, China. A total of 6,589 singleton pregnant

women aged 18–50 years were recruited from four hospitals across three administrative districts (two counties and one county-level city) between March 2020 and March 2022. Participants were enrolled during their first antenatal visit (≥6 weeks of gestation) and followed through delivery until one-week postpartum. Standardized oral glucose tolerance tests (OGTT) were administered at 24–28 gestational weeks for GDM screening, following International Association of Diabetes and Pregnancy Study Groups (IADPSG) criteria. Prospective data collection included: (1) Serial antenatal examinations documenting maternal clinical parameters; (2) Structured face-to-face interviews using validated questionnaires; (3) Biospecimen sampling (blood/urine) at designated timepoints; (4) Systematic recording of perinatal outcomes.

## Ethical considerations

All participants provided written informed consent at enrollment. The study protocol was approved by the Ethics Committee of the Affiliated Hospital of Zunyi Medical University (Approval No.: [2019] H-005), complying with the Declaration of Helsinki principles.

This study employed strict inclusion and exclusion criteria to ensure cohort homogeneity. The inclusion criteria were as follows: pregnant women who had regular prenatal examinations and delivered in these four hospitals; absence of pre-existing comorbidities (excluding diabetes and hypertension), specifically: such as hepatic/renal dysfunction, chronic gastrointestinal disorders, respiratory pathologies, infectious diseases, thyroid dysfunction, and so on; and no current use of medications affecting glucose/lipid metabolism (statins, glucocorticoids, antipsychotics). The exclusion criteria were as follows: missing critical baseline data (>20% variables incomplete) and previous basic diseases. In total, 6,394 women were eventually enrolled on the basis of inclusion and exclusion criteria.

## GDM diagnostic criteria

GDM was diagnosed according to the 2015 American Diabetes Association (ADA) guidelines (*Handelsman et al., 2015*), adopting the International Association of Diabetes and Pregnancy Study Groups (IADPSG)-recommended 75-g 2-hour oral glucose tolerance test (OGTT). The standardized protocol required: administration between 24 and 28 gestational weeks; morning testing following an overnight fast of ≥8 h; venous plasma glucose measurement at three time cut-off points: fasting plasma glucose ≥5.1 mmol/L, 1-h plasma glucose ≥10.0 mmol/l, and 2-h plasma glucose ≥8.5 mmol/l. Diagnosis was confirmed if any single glucose threshold was met or exceeded.

## Assessment of covariates

The participants were interviewed by trained interviewers to obtain social demographic information, including age, sex, pre-pregnancy weight, height, parity, education level, occupation, smoking status, alcohol consumption, number of physical activity frequency per week, family history of diabetes and hypertension, personal history of diabetes and hypertension, *etc*. Moreover, the maternal pregnancy outcomes were investigated, including abortion, premature birth, fetal weight, premature rupture of membranes, placental abruption, fetal asphyxia, intrauterine distress, *etc*.

The measurement of the pre-pregnancy BMI varied due to the differences in adult physique and other aspects. The pre-pregnancy BMI (weight (kg)/height (m)$^2$) was calculated based on the measured height and self-reported pre-pregnancy weight. BMI was categorized according to the 2002 China classification: underweight (BMI < 18.5 kg/m$^2$), normal weight (BMI 18.5–23.9 kg/m$^2$), overweight (BMI 24.0–27.9 kg/m$^2$), and obesity (BMI ≥ 28.0 kg/m$^2$) (*Beifan, 2002*). Pregnancy-induced hypertension (PIH) was diagnosed per ISSHP guidelines when systolic blood pressure (BP) ≥140 mmHg and/or diastolic BP ≥90 mmHg, was first recorded after 20 gestational weeks, confirmed by two measurements ≥4 h apart, without pre-existing hypertension or proteinuria (*Mol et al., 2016*). Preterm birth was defined as delivery occurring between $28^0/7$ and $<36^6/7$ weeks gestation, with gestational age determined by first-trimester ultrasound dating (*Ahumada-Barrios & Alvarado, 2016*). Low birth weight (LBW) was defined as the neonatal weight <2,500 g measured within 1 h of delivery, regardless of gestational age (*FOA, 2017*).

## Statistical methods

EpiData 3.1 (EpiData Association, Odense, Denmark) was used to perform dual-entry verification to ensure data integrity. All data were analyzed using IBM SPSS Statistics version 29.0 (IBM Corp., Armonk, NY, USA). Data were described as mean ± standard deviation (SD) for continuous variables and as frequencies (percentages) for categorical variables. For the univariate analysis, the Pearson $\chi 2$ test or Fisher's test (categorical variables) was used, and independent samples were Student's *t*-test (categorical variables) or Mann–Whitney U test (non-normal distributions). Logistic regression models were used to evaluate associations between socioeconomic indicators (education, occupation, household income) and GDM risk. Binary logistic regression was used to obtain odds ratios (ORs) and 95% confidence intervals (CIs) of SES indicators for GDM, using less than primary school, public sector employees, or household income <30,000 CNY/year as the reference groups.

In the multivariate logistic model, we adjusted for maternal ethnicity, age (continuous), pre-pregnancy BMI (constant), parity (primiparous or multiparous), physical activity before pregnancy (≥3 times/week = 1, 1–2 times/week = 2, less than 1 times/week = 3, few/none = 4), passive smoking, smoking before pregnancy, smoking during pregnancy, hypertension during pregnancy, occupational status, educational level, education level of the husband, occupational status of the husband, household income. We did not adjust for maternal alcohol use before pregnancy and during pregnancy, because only one woman reported no alcohol use before pregnancy, and all women reported alcohol use before pregnancy among 6,394 women included in the analysis.

To evaluate effect modification, we conducted a subgroup analysis to examine the association between SES and gestational diabetes according to maternal age (≤25, 25 to 35 or ≥35 years, the median age of mother at delivery) and prepregnancy BMI (<18.5 or 18.5 to 24 or ≥24 kg/m$^2$, the cut-off value of overweight for China populations). Tests for interaction across subgroups were conducted using the Wald test. Furthermore, we examined the effects of having both higher family monthly income (defined as ≥100,000 CNY/year) and higher education attainment (defined as more than university), and having

either of them for a possible dose–response effect for GDM. To estimate the robustness of our results, we also performed a sensitivity analysis excluding hypertension during pregnancy.

## RESULTS

### Basic information

Of the 5,868 women included in the study, 690 (11.8%) were diagnosed with GDM (Table 1). Compared with non-GDM women, older, with higher pre-pregnancy BMI, government employment, higher income, primiparous, less exercise, and daily passive smoking, history of GDM, diabetes before pregnancy, and hypertension before pregnancy were more likely to develop GDM. They also had a higher probability of developing Pregnancy-induced hypertension (PIH) and had a higher prevalence of both fetal growth restriction (FGR) and macrosomia.

### Risk associations of income and educational attainment with GDM

Neither univariate nor multivariable-adjusted logistic regression models demonstrated significant associations between educational attainment and GDM risk (Table 2, $p > 0.05$ across all comparisons). Compared with women with public sector employees, the ORs of GDM were 0.75 (95% CI [0.56–1.00]), 0.68 (95% CI [0.52–0.88]), and 0.89 (95% CI [0.67–1.18]) for those with private sector employees, freelancers, and farmers in the unadjusted model. After adjustment for potential confounders, the ORs of GDM across categories of occupational status (public sector employees, private sector employees, freelance and farmers) were 1.00 (reference), 0.70 (95% CI [0.48–1.00]), 0.65 (95% CI [0.45–0.94]) and 0.72 (95% CI [0.48–1.07]).

In the univariate model and multivariate-adjusted models, household income was positively associated with the GDM risk (Table 2). In the univariate model, the OR of GDM for different income level categories was 1.00 (reference), 1.30 (95% CI [0.94–1.79]), 1.97 (95% CI [1.46–2.66]), 2.11 (95% CI [1.52–2.92]), and 2.72 (95% CI [1.34–5.53]), respectively. After adjusting for potential confounders, the OR for GDM across income level categories was 1.00 (reference), 1.04 (95% CI [0.73–1.49]), 1.91 (95% CI [1.39–2.62]), 1.72 (95% CI [1.21–2.45]), 2.54 (95% CI [1.22–5.30]), respectively.

### GDM is associated with SES categories, stratified by maternal age

We performed subgroup analysis according to maternal age ($\leq 25$, 25–35, and $\geq 35$ years) and pre-pregnancy BMI ($<18.5$, 18.5–24, and $\geq 24$ kg/m$^2$)(Tables 3 and 4). There were no interactions between the mother's education level, occupational status, household income, and maternal age, and pre-pregnancy BMI on gestational diabetes risk ($P$ for interaction $> 0.05$).

The results showed that the mother's education level under 35 years old was not associated with the GDM risk in the univariate and model-adjusted models. There was no significant association between educational level and GDM observed in the univariate model. However, a higher educational level was associated with a lower risk of gestational diabetes in multivariate models (primary school or below *vs.* undergraduate: OR, 0.14 (95% CI [0.03–0.73]).

**Table 1 Characteristics of participants with gestational diabetes and non-gestational.**

|  | Non-gestational diabetes (5,178) | Gestational diabetes ($n = 690$) | P value |
|---|---|---|---|
| **Maternal age (years)** | 25.81 ± 6.14 | 27.09 ± 6.54 | **<0.0001**[*] |
| ≤25 | 2,455 (47.2) | 275 (39.9) | |
| 25–35 | 2,375 (45.9) | 332 (48.1) | |
| ≥35 | 358 (6.9) | 83 (12.0) | |
| **Pregnancy BMI (kg/m2)** | 21.88 ± 3.49 | 23.04 ± 3.94 | **<0.0001**[*] |
| <18.5 | 665 (12.8) | 65 (9.4) | |
| 18.5–24 | 3,369 (65.1) | 395 (57.2) | |
| ≥24 | 1,144 (22.1) | 230 (33.3) | |
| Ethnic | | | 0.051 |
| Han | 4,963 (95.8) | 672 (97.4) | |
| Others | 215 (4.2) | 18 (2.6) | |
| **Educational level** | | | 0.783 |
| Primary school or below | 341 (6.6) | 50 (7.2) | |
| Junior middle school | 2,353 (45.4) | 319 (46.2) | |
| Enior school | 1,304 (25.2) | 174 (25.2) | |
| Undergraduate | 1,180 (22.8) | 147 (21.3) | |
| **Occupational status** | | | **0.005**[*] |
| Public officials | 494 (9.5) | 86 (12.5) | |
| Private sector employees | 968 (18.7) | 126 (18.3) | |
| Freelance | 2,661 (51.4) | 314 (45.5) | |
| Farmers | 1,055 (20.4) | 164 (23.8) | |
| **Household income** | | | **<0.0001**[*] |
| <30,000 yuan/year (lowest) | 682 (13.2) | 54 (7.8) | |
| 30,000–99,999 yuan/year | 1,463 (28.3) | 150 (21.7) | |
| 100,000–149,999 yuan/year | 2,095 (40.5) | 327 (47.4) | |
| 150,000–249,999 yuan/year | 887 (17.1) | 148 (21.4) | |
| ≥250,000 yuan/year (highest) | 51 (1.0) | 11 (1.6) | |
| **Smoking before pregnancy, n (%)** | | | 0.978 |
| Yes | 97 (1.9) | 13 (1.9) | |
| No | 5,071 (98.1) | 674 (98.1) | |
| **Smoking during pregnancy, n (%)** | | | 0.101 |
| Yes | 4 (0.1) | 2 (0.3) | |
| No | 5,174 (99.9) | 688 (99.7) | |
| **Alcohol use before pregnancy, n (%)** | | | 0.146 |
| Yes | 151 (3.4) | 14 (2.3) | |
| No | 4,239 (96.6) | 591 (97.7) | |
| **Alcohol use during pregnancy, n (%)** | | | |
| Yes | NA | NA | |
| No | 5,177 | 690 | |
| **Passive smoking, n (%)** | | | **0.003**[*] |
| Yes | 1,524 (29.4) | 241 (34.9) | |
| No | 3,654 (70.6) | 499 (65.1) | |

| | Non-gestational diabetes (5,178) | Gestational diabetes (*n* = 690) | *P* value |
|---|---|---|---|
| **Physical activity, *n* (%)** | | | **0.028**[*] |
| Never or rarely | 1,063 (20.6) | 172 (25.0) | |
| 1–2 days/week | 316 (6.1) | 47 (6.8) | |
| 3–4 days/week | 343 (6.6) | 37 (5.4) | |
| More 5 days/week | 3,443 (66.7) | 431 (62.7) | |
| **Parity, *n* (%)** | | | **0.200** |
| Primiparous | 3,116 (60.2) | 447 (64.8) | |
| Multiparous | 2,062 (39.8) | 243 (35.2) | |
| **History of GDM, *n* (%)** | | | **<0.0001**[*] |
| Yes | 22 (0.4) | 31 (4.5) | |
| No | 5,156 (99.6) | 659 (95.5) | |
| **Diabetes before pregnancy, *n* (%)** | | | **<0.0001**[*] |
| Yes | 33 (0.6) | 79 (11.4) | |
| No | 5,145 (99.4) | 611 (88.6) | |
| **Hypertension before pregnancy, *n* (%)** | | | **<0.0001**[*] |
| Yes | 53 (1.0) | 24 (3.5) | |
| No | 5,125 (99.0) | 666 (96.5) | |
| **Thyroid disorders before pregnancy, *n* (%)** | | | **0.5120** |
| Yes | 129 (2.5) | 20 (2.9) | |
| No | 5,049 (97.5) | 670 (97.1) | |
| **Hypertension during pregnancy, *n* (%)** | | | **<0.0001**[*] |
| Yes | 222 (4.3) | 96 (13.9) | |
| No | 4,956 (95.7) | (594 (86.1) | |
| **Preterm birth, *n* (%)** | | | |
| Yes | 286 (5.5) | 71 (10.3) | **<0.0001**[*] |
| No | 4,892 (94.5) | 619 (89.7) | |
| **Birth weight, *n* (%)** | | | **<0.0001**[*] |
| Low birth weight | 51 (1.0) | 9 (1.3) | |
| Normal birth weight | 5,002 (96.6) | 646 (93.6) | |
| Macrosomia | 125 (2.4) | 35 (5.1) | |

**Notes.**
[*]*P* < 0.05, the difference is statistically significant.
Boldface: the difference was statistically significant.

The mother's occupational status under 25 years old was associated with the GDM risk in the univariate analysis. Compared with women with public sector employees, the ORs of GDM were 0.77 (95% CI [0.53–1.11]), 0.60 (95% CI [0.44–0.83]), and 0.96 (95% CI [0.67–1.38]) for those with private sector employees, freelancers, and farmers in the unadjusted model. However, no significant association between occupational status and GDM was observed in the multivariate-adjusted model. There was no significant association between occupational status and GDM observed in the univariate model and multivariate-adjusted model when the mother was 25–35 years old. The mother's occupational status above 35 years old was associated with the GDM risk in the univariate analysis. Compared with women with public sector employees, the ORs of GDM were 0.38
**Table 2   Odds ratios (ORs) and 95% confidence intervals for gestational diabetes according to socio-economic status categories.**

| Socio-economic variables | No. of participants | No. of cases | Unadjusted OR | Adjusted OR# |
|---|---|---|---|---|
| **Educational level** | | | 0.783 | 0.400 |
| Primary school or below | 50 (7.2) | 341 (6.6) | 1.00 (reference) | 1.00 (reference) |
| Junior middle school | 319 (46.2) | 2,353 (45.4) | 0.93 (0.67, 1.27) | 1.17 (0.80, 1.69) |
| Enior school | 174 (25.2) | 1,304 (25.2) | 0.91 (0.65, 1.27) | 1.10 (0.75, 1.61) |
| Undergraduate | 147 (21.3) | 1,180 (22.8) | 0.85 (0.60, 1.20) | 0.91 (0.60, 1.37) |
| **Occupational status** | | | **0.005***  | 0.131 |
| Public officials | 86 (12.5) | 494 (9.5) | 1.00 (reference) | 1.00 (reference) |
| Private sector employees | 126 (18.3) | 968 (18.7) | 0.75 (0.56, 1.00) | 0.70 (0.48, 1.00) |
| Freelance | 314 (45.5) | 2,661 (51.4) | **0.68 (0.52, 0.88)** | **0.65 (0.45, 0.94)** |
| Farmers | 164 (23.8) | 1,055 (20.4) | 0.89 (0.67, 1.18) | 0.72 (0.48, 1.07) |
| **Household income** | | | **<0.0001*** | **<0.0001*** |
| <30,000 yuan/year (lowest) | 682 (13.2) | 54 (7.8) | 1.00 (reference) | 1.00 (reference) |
| 30,000–99,999 yuan/year | 1,463 (28.3) | 150 (21.7) | 1.30 (0.94, 1.79) | 1.04 (0.73, 1.49) |
| 100,000–149,999 yuan/year | 2,095 (40.5) | 327 (47.4) | **1.97 (1.46, 2.66)** | **1.91 (1.39, 2.62)** |
| 150,000–249,999 yuan/year | 887 (17.1) | 148 (21.4) | **2.11 (1.52, 2.92)** | **1.72 (1.21, 2.45)** |
| ≥25 yuan/year (highest) | 51 (1.0) | 11 (1.6) | **2.72 (1.34, 5.53)** | **2.54 (1.22, 5.30)** |

Notes.

#Adjusted for maternal ethnic, age, pre-pregnancy BMI, parity, physical activity, passive smoking, smoking before pregnancy, smoking during pregnancy, hypertension during pregnancy, pre-pregnancy hypertension, diabetes mellitus, and gestational diabetes history, occupational status, educational level, education level of the husband, occupational status of the husband, household income, except for itself.

*$P < 0.05$, The difference is statistically significant.

Boldface: the difference was statistically significant.

(95% CI [0.15–0.91]) for those with farmers in the unadjusted model. Compared with public sector employees, private sector employees, freelance, and farmers are protective factors for the occurrence of GDM in the adjusted model; the ORs of GDM were 0.18 (95% CI [0.05–0.70]), 0.16 (95% CI [0.04–0.0.66]) and, 0.12 (95% CI [0.03–0.52]).

In the univariate model and multivariate-adjusted models, household income was positively associated with the GDM risk at maternal age under 35 years. Within a certain range of household income (<250,000 CNY/year). Higher income (>250,000 CNY/year) increased the risk of gestational diabetes gestational diabetes in mothers over 35 years old in the unadjusted model, but the multivariate-adjusted model showed that the risk of gestational diabetes in mothers over 35 years of age was independent of household income.

## GDM is associated with SES categories, stratified by BMI

No significant associations were observed between educational levels and GDM risk across all BMI strata in both unadjusted ($p$ range: 0.676−0.960) and adjusted analyses (aOR range: 0.60–1.33, $p > 0.05$). Compared with government employees, freelancers were relatively less likely to have GDM in the univariate and multivariate models, with a BMI <18.5 kg/m$^2$. Occupation was not associated with a BMI in the univariate and multivariate-adjusted models when BMI was ≥18.5 kg/m$^2$. In the univariate and multivariate-adjusted models, household income was positively associated with the GDM risk at maternal BMI over 18.5 kg/m$^2$.

**Table 3 Odds ratios (ORs) of gestational diabetes associated with socio-economic status categories, stratified by maternal age.**

| | ≤25 years (n =) | | 25–35 years (n =) | | ≥35 years (n =) | | P for Interaction |
|---|---|---|---|---|---|---|---|
| Socio-economic variables | Unadjusted OR | Adjusted OR[#] | Unadjusted OR | Adjusted OR[#] | Unadjusted OR | Adjusted OR[#] | |
| Educational level | 0.782 | 0.765 | 0.493 | 0.635 | 0.492 | **0.031**[*] | 0.456 |
| Primary school or below | 1.00 (reference) | 1.00 (reference) | 1.00 (reference) | 1.00 (reference) | 1.00 (reference) | 1.00 (reference) | |
| Junior middle school | 1.03 (0.65, 1.65) | 1.31 (0.67, 2.55) | 0.94 (0.49, 1.81) | 1.06 (0.57, 1.97) | 0.87 (0.45, 1.66) | 1.32 (0.60, 2.89) | |
| Enior school | 0.98 (0.60, 1.59) | 1.25 (0.66, 2.37) | 0.94 (0.48, 1.85) | 0.95 (0.50, 1.76) | 1.29 (0.61, 2.71) | 1.66 (0.63, 4.35) | |
| Undergraduate | 1.15 (0.71, 1.86) | 1.43 (0.72, 2.80) | 0.73 (0.36, 1.47) | 0.82 (0.43, 1.57) | 0.70 (0.29, 1.72) | **0.22 (0.06, 0.87)** | |
| Occupational status | **0.003**[*] | 0.466 | 0.382 | 0.483 | 0.193 | **0.039**[*] | 0.391 |
| Public officials | 1.00 (reference) | 1.00 (reference) | 1.00 (reference) | 1.00 (reference) | 1.00 (reference) | 1.00 (reference) | |
| Private sector employees | 0.77 (0.53, 1.11) | 0.98 (0.58, 1.64) | 0.67 (0.31, 1.44) | 0.63 (0.35, 1.15) | 0.49 (0.19, 1.22) | **0.18 (0.05, 0.70)** | |
| Freelance | **0.60 (0.44, 0.83)** | 0.77 (0.45, 1.29) | 0.65 (0.31, 1.36) | 0.66 (0.36, 1.21) | 0.52 (0.23, 1.13) | **0.16 (0.04, 0.66)** | |
| Farmers | 0.96 (0.67, 1.38) | 0.98 (0.55, 1.74) | 0.81 (0.38, 1.72) | 0.70 (0.37, 1.34) | **0.38 (0.15, 0.91)** | **0.12 (0.03, 0.52)** | |
| Household income | **<0.0001**[*] | **<0.0001**[*] | **0.002**[*] | **<0.0001**[*] | 0.273 | 0.063 | 0.119 |
| <30,000 yuan/year (lowest) | 1.00 (reference) | 1.00 (reference) | 1.00 (reference) | 1.00 (reference) | 1.00 (reference) | 1.00 (reference) | |
| 30,000–99,999 yuan/year | 0.94 (0.60, 1.49) | 0.82 (0.47, 1.43) | 1.33 (0.72, 2.46) | 0.99 (0.56, 1.78) | 2.13 (0.84, 5.44) | 2.47 (0.80, 7.69) | |
| 100,000–149,999 yuan/year | **1.97 (1.35, 2.88)** | **1.89 (1.23, 2.91)** | **2.07 (1.14, 3.75)** | **2.07 (1.21, 3.55)** | 1.61 (0.62, 4.16) | 2.04 (0.68, 6.15) | |
| 150,000–249,999 yuan/year | **2.05 (1.34, 3.12)** | **1.65 (1.01, 2.69)** | **2.24 (1.19, 4.20)** | **2.11 (1.18, 3.77)** | 1.78 (0.65, 4.88) | 0.83 (0.23, 2.96) | |
| ≥250,000 yuan/year (highest) | 1.07 (0.24, 4.70) | 1.37 (0.30, 6.29) | **3.52 (1.22, 10.22)** | 2.67 (0.93, 7.71) | **5.75 (1.03, 32.17)** | 7.20 (0.95, 54.60) | |

**Notes.**
[#]Adjusted for maternal ethnic, age, pre-pregnancy BMI, parity, physical activity, passive smoking, smoking before pregnancy, smoking during pregnancy, hypertension during pregnancy, pre-pregnancy hypertension, diabetes mellitus, and gestational diabetes history, occupational status, educational level, education level of the husband, occupational status of the husband, household income, except for itself.
[*]$P < 0.05$, The difference is statistically significant.
Boldface: the difference was statistically significant.

## Integrated effects of SES for GDM

Using more than a university education as the higher education level, the middle-high income and high income as the higher income. The ORs for both were 1.56 (95% CI [1.12–2.17]), the ORs for higher household income or higher education level were 1.21 (95% CI [0.97–1.51]) and 0.80 (95% CI [0.64–1.01]), respectively with having none of them as the reference category ($P = 0.002$, Table 5, model 1). Adjustment for maternal age, pre-pregnancy BMI, hypertension during pregnancy, passive smoking, and physical activity based on modle1 did not substantially increase or decline the effect sizes of having both factors and having either of them, the ORs for both were 1.47 (95% CI [1.03–2.09]), the ORs for higher household income or higher education level were 1.07 (95% CI [0.84–1.37]) and 0.83 (95% CI [0.65–1.06]), respectively with having none of them as the reference category ($P = 0.039$ Table 5, model 2). In the same, adjustment for occupational status based on modle2 did substantially declined the effect sizes of having both factors and having either of them OR = 1.14 (95% CI [0.90–1.43]), 0.67 (95% CI [0.51–0.95]) and 1.23 (95% CI [0.84–1.78]) ($P = 0.001$, Table 5, model 4). In the same, adjustment for occupational status based on modle2 did substantially declined the effect sizes of having both factors and having either of them OR = 1.03 (95% CI [0.80–1.32]), 0.72 (95% CI [0.54–0.95]) & 1.22 (95% CI [0.82–1.81]), ($P = 0.028$, Table 5, model 4).

**Table 4  Odds ratios (ORs) of gestational diabetes associated with socio-economic status categories, stratified by BMI.**

| | <18.5 kg/m² (n = 799) | | 18.5–24 kg/m² (n =) | | ≥24 kg/m² (n =) | | P for Interaction |
|---|---|---|---|---|---|---|---|
| Socio-economic variables | Unadjusted OR | Adjusted OR* | Unadjusted OR | Adjusted OR* | Unadjusted OR | Adjusted OR* | |
| **Educational level** | 0.791 | 0.727 | 0.960 | 0.676 | 0.741 | 0.609 | 0.900 |
| Primary school or below | 1.00 (reference) | 1.00 (reference) | 1.00 (reference) | 1.00 (reference) | 1.00 (reference) | 1.00 (reference) | |
| Junior middle school | 0.76 (0.25, 2.31) | 1.20 (0.26, 5.41) | 0.92 (0.60, 1.41) | 1.12 (0.68, 1.84) | 1.03 (0.60, 1.78) | 1.25 (0.66, 2.39) | |
| Enior school | 0.60 (0.19, 1.93) | 0.92 (0.21, 3.98) | 0.89 (0.57, 1.40) | 1.00 (0.60, 1.67) | 1.16 (0.66, 2.06 | 1.33 (0.69, 2.55) | |
| Undergraduate | 0.79 (0.25, 2.51) | 0.69 (0.15, 3.20) | 0.89 (0.57, 1.40) | 0.90 (0.53, 1.52) | 0.91 (0.49, 1.68) | 0.94 (0.45, 1.95) | |
| **Occupational status** | 0.145 | 0.092 | 0.054 | 0.483 | 0.263 | 0.705 | 0.684 |
| Public officials | 1.00 (reference) | 1.00 (reference) | 1.00 (reference) | 1.00 (reference) | 1.00 (reference) | 1.00 (reference) | |
| Private sector employees | 0.57 (0.25, 1.30) | 0.39 (0.12, 1.23) | 0.87 (0.59, 1.29) | 0.78 (0.49, 1.24) | 0.62 (0.36, 1.07) | 0.69 (0.35, 1.37) | |
| Freelance | **0.42 (0.20, 0.87)** | **0.22 (0.07, 0.72)** | 0.72 (0.51, 1.02) | 0.70 (0.43, 1.13) | 0.69 (0.43, 1.10) | 0.85 (0.43, 1.68) | |
| Farmers | 0.53 (0.23, 1.21) | **0.26 (0.07, 0.95)** | 1.01 (0.69, 1.47) | 0.79 (0.47, 1.33) | 0.82 (0.50, 1.37) | 0.87 (0.42, 1.78) | |
| **Household income** | 0.431 | 0.449 | <0.0001* | <0.0001* | <0.0001* | <0.015* | 0.318 |
| <30,000 yuan/year (lowest) | 1.00 (reference) | 1.00 (reference) | 1.00 (reference) | 1.00 (reference) | 1.00 (reference) | 1.00 (reference) | |
| 30,000–99,999 yuan/year | 0.75 (0.30, 1.87) | 0.73 (0.24, 2.18) | 1.16 (0.75, 1.77) | 0.87 (0.54, 1.39) | 1.75 (0.95, 3.23) | 1.58 (0.81, 3.08) | |
| 100,000–149,999 yuan/year | 1.15 (0.51, 2.60) | 0.97 (0.40, 2.37) | **2.09 (1.42, 3.09)** | **1.96 (1.30, 2.96)** | **2.06 (1.14, 3.73)** | **2.26 (1.21, 4.21)** | |
| 150,000–249,999 yuan/year | 0.99 (0.38, 2.58) | 0.54 (0.18, 1.64) | **2.34 (1.54, 3.57)** | **2.08 (1.33, 3.25)** | 2.13 (1.12, 4.03) | 1.64 (0.82, 3.29) | |
| ≥25 yuan/year (highest) | 3.46 (0.60, 20.04) | 2.69 (0.42, 17.24) | 1.51 (0.44, 5.24) | 1.15 (0.31, 4.23) | **3.51 (1.18, 10.42)** | **4.87 (1.55, 15.30)** | |

**Notes.**

*$P < 0.05$, The difference is statistically significant.

# Adjusted for maternal ethnic, age, pre-pregnancy BMI, parity, physical activity, passive smoking, smoking before pregnancy, smoking during pregnancy, hypertension during pregnancy, pre-pregnancy hypertension, diabetes mellitus, and gestational diabetes history, occupational status, educational level, education level of the husband, occupational status of the husband, household income, except for itself.

Boldface: the difference was statistically significant.

# DISCUSSION

Although numerous studies have investigated the association between SES and GDM, the results remain inconclusive. Existing literature suggests that higher SES may correlate with reduced GDM risk, potentially attributable to improved access to nutritional resources and healthcare services. Conversely, others indicate that this relationship could be mediated by contextual factors, including regional culture, pre-pregnancy BMI, and dietary patterns. In this study, we conducted a systematic evaluation of the SES and GDM relationship within the unique sociocultural context of Zunyi, China. After adjusting for potential confounders, After adjusting for potential confounders, our analysis revealed three key findings: (1) Advanced educational attainmentwas associated with a reduced GDM risk specifically among women aged >35 years; (2) Private sector workers, freelancers, and agricultural laborers exhibited lower GDM risk compared to public sector employees; (3) Household income demonstrated a significant positive association with the prevalence of GDM.

In general, higher education attainment is associated with enhanced health literacy and health-promoting behavior. Pregnant women with advanced education demonstrate greater awareness of GDM risk factors and stronger adherence to preventive lifestyle modifications, including balanced nutritional intake and regular physical activity, which collectively mitigate GDM risk. This pattern is corroborated by multicenter studies:

**Table 5 Multivariable odds ratios of income and education attainment for GDM.**

|  | B | S.E. | Wald | OR (95% CI) | P |
|---|---|---|---|---|---|
| Model 1 |  |  | 15.072 |  | **0.002** |
| Lower income and lower education |  |  |  | 1.00 (reference) | 1.00 |
| Higher income and lower education | 0.189 | 0.114 | 2.743 | 1.21 (0.97, 1.51) | 0.098 |
| Lower income and higher education | −0.219 | 0.117 | 3.499 | 0.80 (0.64, 1.01) | 0.061 |
| Higher income and higher education | 0.445 | 0.168 | 6.998 | **1.56 (1.12, 2.17)** | **0.008** |
| Model 2 |  |  | 8.371 |  | **0.039** |
| Lower income and lower education |  |  |  | 1.00 (reference) |  |
| Higher income and lower education | 0.065 | 0.125 | 0.272 | 1.07 (0.84, 1.37) | 0.602 |
| Lower income and higher education | −0.188 | 0.124 | 2.300 | 0.83 (0.65, 1.06) | 0.129 |
| Higher income and higher education | 0.385 | 0.179 | 4.618 | **1.47 (1.03, 2.09)** | **0.032** |
| Model 3 |  |  | 15.559 |  | **0.001** |
| Lower income and lower education |  |  |  | 1.00 (reference) |  |
| Higher income and lower education | 0.127 | 0.117 | 1.184 | 1.14 (0.90, 1.43) | 0.277 |
| Lower income and higher education | −0.404 | 0.139 | 8.429 | **0.67 (0.51, 0.88)** | **0.004** |
| Higher income and higher education | 0.203 | 0.191 | 1.126 | 1.23 (0.84, 1.78) | 0.289 |
| Model 4 |  |  | 9.092 |  | **0.028** |
| Lower income and lower education |  |  |  | 1.00 (reference) |  |
| Higher income and lower education | 0.028 | 0.127 | 0.048 | 1.03 (0.80, 1.32) | 0.827 |
| Lower income and higher education | −0.335 | 0.147 | 5.194 | **0.72 (0.54, 0.95)** | **0.023** |
| Higher income and higher education | 0.199 | 0.201 | 0.978 | 1.22 (0.82, 1.81) | 0.323 |

**Notes.**
Model 1: Unadjusted.
Model 2: Adjusted for maternal age, pre-pregnancy BMI, hypertension during pregnancy, passive smoking, physical activity.
Model 3: Adjusted for occupational status;
Model 4: Adjusted for maternal age, pre-pregnancy BMI, hypertension during pregnancy, pre-pregnancy hypertension, diabetes mellitus, and gestational diabetes history, passive smoking, physical activity, occupational status.
Boldface: the difference was statistically significant.

Research in Tianjin, China, revealed that pregnant women with higher education had a significantly lower GDM risks compared to those with high school education or less (*Leng et al., 2017*); A Finnish study of 5,962 pregnant mothers found that the prevalence of GDM was inversely related to education, suggesting enhanced knowledge of GDM prevention among more educated women (*Roustaei et al., 2023*). Another investigation found that 49.1% of postgraduate-educated participants possessed excellent knowledge of GDM, compared to 20.9% of those with only basic schooling (*Almatrafi & Sekhar, 2024*), further emphasizing education's role in gestational diabetes awareness. Our analysis indicates that after adjusting for potential confounders, higher education levels were associated with reduced GDM risk exclusively in women aged ≥35 years, while no such association existed in younger age groups. This discrepancy may relate to age-related differences in the lifestyle patterns, dietary habits,and health information processing capabilities. Younger pregnant women may preferentially rely on familial and social networks for health information rather than formal education-based knowledge (*Yingni et al., 2023*).

Women ≥35 years typically occupy established career positions with stable incomes and socioeconomic status, granting greater resources and time for health management.

Additionally, advancing age often corresponds with heightened health consciousness and proactive health behaviors. Also, these finding emphasizes the synergistic influence of age and education on GDM risk and suggest the need for age-tailored preventive strategies for pregnant women. These findings highlight the synergistic influence of age and education on GDM risk and suggest the need for age-tailored preventive strategies for pregnant women.

The study on household income and the GDM risk revealed a positive correlation, with GDM incidence rising as income increased—a finding inconsistent with existing literature. For instance, research in Wuhan, China, from September 2012 to October 2014, found that higher household incomes did not necessarily lead to an increased GDM risk (*Li et al., 2018*). Similarly, a 2010–2012 Tianjin study reported reduced GDM risk among middle- and high-income groups (*Leng et al., 2017*), while Finnish data demonstrated declining GDM rates with higher pre-tax income among primiparous women (*Roustaei et al., 2023*). These studies, conducted in economically developed regions, suggest that higher-income populations may benefit from better living conditions, greater access to health resources (*e.g.*, regular exercise opportunities and balanced diets), and potentially higher educational attainment, which collectively promote healthier lifestyles and lower GDM risk. However, this phenomenon is not universal. A study in Shenzhen showed that the incidence of GDM was 14.7% in pregnant women with high family income, significantly exceeding rates in middle (8.0%) and low-income (4.0%) groups (*Jiazhang, Hongmin & Wei, 2017*). Our observations align with these findings, suggesting in prevalence with higher incidence among affluent households. This pattern may stem from enhanced healthcare accessibility and greater health consciousness, enabling more frequent GDM diagnoses in high-income populations (*Leng et al., 2017*). Secondly, elevated occupational stress and demanding workloads characteristic of affluent demographics may contribute to physical inactivity patterns through fast-paced lifestyles and irregular eating habits (*Zhao et al., 2021*). Concurrently, chronic workplace pressures may elevate psychological stress levels, potentially disrupting endocrine system function and thereby increasing gestational diabetes susceptibility (*Zheng, 2023*). Furthermore, heightened focus on nutrient-dense diets in affluent environments might paradoxically lead to overconsumption of calorie-dense, lipid-rich foods, promoting weight gain and insulin resistance, established risk factors for GDM. These complex interactions underscore the necessity of comprehensive analysis when examining income-GDM associations, requiring simultaneous consideration of geographical variations, socioeconomic determinants, behavioral patterns, and nutritional practices. Subsequent research should prioritize elucidating the mechanistic pathways through which these multidimensional factors influence GDM development, thereby informing more targeted preventive interventions.

Current epidemiological investigations reveal occupation-dependent variations in GDM susceptibility among pregnant populations. Our analysis demonstrates that private sector employees, self-employed individuals, and agricultural workers exhibit reduced GDM risk relative to their public sector counterparts. Notably, BMI-stratified analysis identified amplified occupational influence on GDM development in underweight gravidas (BMI <18.5 kg/m$^2$). Differential occupational energy expenditure patterns emerge as a pivotal determinant, with physically demanding roles characteristic of commercial,

entrepreneurial, and agrarian occupations potentially conferring metabolic protection through enhanced caloric utilization (*Leng et al., 2016*). Mechanistically, occupation-mediated physical activity may modulate GDM pathogenesis by maintaining optimal body composition and glycemic homeostasis (*Leng et al., 2016*). Conversely, public sector occupations—particularly administrative roles predominantly held by females—frequently involve prolonged sedentary behaviors, a recognized risk amplifier supported by clinical evidence linking structured exercise regimens (including brisk walking) to GDM risk attenuation (*Padayachee & Coombes, 2015*; *Nan et al., 2022*). This aligns with WHO guidelines advocating systematic replacement of sedentary time with purposeful movement across all life stages, including antenatal and postnatal periods, to optimize metabolic health outcomes (*Scheuer et al., 2023*).

The integrated analysis revealed a paradoxical risk profile wherein elevated SES correlated with heightened GDM susceptibility. However, this pattern contrasts with findings from the 2010 to 2012 Tianjin cohort demonstrating inverse SES-GDM associations, potentially attributable to regional health policy disparities. As a metropolitan hub adjacent to the national capital, Tianjin's geopolitical advantages facilitate superior healthcare access, further reinforced by its pioneering implementation of structured GDM prevention frameworks through metropolitan tertiary antenatal networks since 1998 (*Liu et al., 2018*). Crucially, occupational parameter adjustment attenuated SES-related GDM risk differentials, suggesting confounding mediation through employment characteristics. The protective mechanism in educated cohorts may involve enhanced pregnancy weight management capabilities and optimized glycemic regulation through improved health literacy and prenatal self-management practices (*Rönö et al., 2019*).

One of the strengths of our study is that the diagnosis of GDM was confirmed through a standardized oral glucose tolerance test (OGTT) rather than relying on self-reported data, thereby minimizing the risk of diagnostic misclassification. Additionally, a large amount of covariate data was collected through face-to-face interviews, medical records, and medical measurements, allowing the researchers to adjust for potential confounders of GDM. Furthermore, this study used a prospective cohort design with a relatively large sample size, which increases the reliability and representativeness of the statistical results. Of course, our studies also have some limitations. Firstly, information on household income is obtained through face-to-face questionnaires, which may introduce bias. Because household income is private in China, some people are reluctant to tell others, and income can fluctuate over time, making it more likely to be misclassified. Secondly, pre-pregnancy BMI was estimated using weight and height at the first antenatal visit rather than pre-pregnancy data. Although pregnant women gain very little weight in the first trimester of pregnancy, this measurement may still be biased. In addition, the study participants were from Zunyi, Guizhou Province, and may not be representative of other regions due to the specificity of Zunyi's geographical location and economic situation.

## CONCLUSIONS

In conclusion, there is a correlation between the socioeconomic status of pregnant women and their GDM risk. We demonstrated that, after adjusting for potential confounders,

higher educational attainment is associated with a reduced GDM risk in women aged $\geq 35$ years. Private sector workers, freelancers, and agricultural laborers are associated with a lower GDM risk compared with public employees. Household income was significantly and positively associated with the prevalence of GDM. To provide an in-depth understanding of the relationship between SES and the risk of developing GDM and its underlying mechanisms, and to provide evidence for new prevention and intervention strategies for GDM and insulin resistance. Raising public awareness of GDM, especially for fertile women, is a key measure to reduce the incidence of GDM. In addition, the government and healthcare institutions should consider providing more health education and counseling services to help high-risk groups better manage their GDM.

## ACKNOWLEDGEMENTS

The People's Hospital of Xishui County and the People's Hospital of Meitan County helped to collect data.

### Funding

This work was supported by Nature Science Foundation of Guizhou Provincial (QKH-J[2022]YB612), National Key R&D Program of China (2018YFC1004300, 2018YFC1004302), Science & Technology Program of Guizhou Province, China (QKHHBZ [2020]3002), City School Joint Foundation Project (QKH-PTRC[2020]-018 & ZYKH-HZ-Z[2021]292), City School Joint Foundation Project (QKH-PTRC[2019]-003 & ZYKH-HZ-Z[2020]64), Start-up Foundation for Doctors of Zunyi Medical University (QKH-PTRC[2019]-032), Natural Science Foundation of Guizhou Province (QKH-J[2020]1Y358), Guizhou Provincial Education Reform Project (SJJG2022-02-166), High-level innovative talents in Guizhou Province (GCC[2022]039-1). The funders had no role in study design, data collection and analysis, decision to publish, or preparation of the manuscript.

### Grant Disclosures

The following grant information was disclosed by the authors:
Nature Science Foundation of Guizhou Provincial: QKH-J[2022]YB612.
National Key R&D Program of China: 2018YFC1004300, 2018YFC1004302.
Science & Technology Program of Guizhou Province, China: QKHHBZ [2020]3002.
City School Joint Foundation Project: QKH-PTRC[2020]-018, ZYKH-HZ-Z[2021]292.
City School Joint Foundation Project: QKH-PTRC[2019]-003, ZYKH-HZ-Z[2020]64.
Start-up Foundation for Doctors of Zunyi Medical University: QKH-PTRC[2019]-032.
Natural Science Foundation of Guizhou Province: QKH-J[2020]1Y358.
Guizhou Provincial Education Reform Project: SJJG2022-02-166.
High-level innovative talents in Guizhou Province: GCC[2022]039-1.

### Competing Interests

The authors declare there are no competing interests.

## Author Contributions

- Yan Xie conceived and designed the experiments, performed the experiments, analyzed the data, prepared figures and/or tables, authored or reviewed drafts of the article, and approved the final draft.
- Jing Jiang conceived and designed the experiments, performed the experiments, analyzed the data, prepared figures and/or tables, authored or reviewed drafts of the article, and approved the final draft.
- Jing Yang conceived and designed the experiments, prepared figures and/or tables, and approved the final draft.
- Shiyi Gao conceived and designed the experiments, prepared figures and/or tables, and approved the final draft.
- Rong Zeng performed the experiments, authored or reviewed drafts of the article, and approved the final draft.
- Yijun Liu analyzed the data, authored or reviewed drafts of the article, and approved the final draft.
- LingLu Wang analyzed the data, authored or reviewed drafts of the article, and approved the final draft.
- Pei Xu performed the experiments, prepared figures and/or tables, and approved the final draft.
- Kunming Tian analyzed the data, prepared figures and/or tables, and approved the final draft.
- Shimin Xiong performed the experiments, prepared figures and/or tables, and approved the final draft.
- Xingyan Liu conceived and designed the experiments, prepared figures and/or tables, and approved the final draft.
- Xubo Shen conceived and designed the experiments, authored or reviewed drafts of the article, and approved the final draft.
- Hong Pan conceived and designed the experiments, authored or reviewed drafts of the article, and approved the final draft.
- YuanZhong Zhou conceived and designed the experiments, authored or reviewed drafts of the article, and approved the final draft.

## Human Ethics

The following information was supplied relating to ethical approvals (*i.e.*, approving body and any reference numbers):

Each participant received a detailed description of the study procedures and provided written informed consent at euponoenrollmentis study was ethically reviewed by the Affiliated Hospital of Zunyi Medical University (batch No.:[2019] H-005).

## Data Availability

The raw data are available in the Supplementary File.

## Supplemental Information

Supplemental information for this article can be found online at http://dx.doi.org/10.7717/peerj.19782#supplemental-information.

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
