# Peer review of "Socioeconomic status and risk of gestational diabetes mellitus among pregnant women in Zunyi, China"

_PeerJ, doi:10.7717/peerj.19782_

## Round 0.1 · original submission · Major Revisions

Dear Authors,

Your study tackles an important question, but there are critical gaps that need attention. The framing lacks a strong justification, the methodology is not entirely transparent, and key variables are insufficiently explored. The results raise questions about the robustness of statistical models and the independence of socio-economic status as a risk factor.

The discussion would benefit from a clearer structure, engaging more directly with the study’s limitations and contextual significance. The conclusions should more accurately reflect the data. Additionally, the clarity of writing needs improvement, and a statistician’s input could enhance the validity of findings.

·

Basic reporting

The study is well-designed, with a sufficiently large sample size that aligns with the research hypothesis.

Experimental design

The results are relatively well-reasoned and comprehensive.

Validity of the findings

However, additional analyses are needed to enhance the study's depth and robustness.

Additional comments

This is an interesting study on the association and significance of socio-economic status and the risk of gestational diabetes mellitus among pregnant women. However, I have the following suggestions:
- Some variables related to the general characteristics of gestational diabetes mellitus, such as the time of diagnosis, G0 and G1 indices, waist-to-hip ratio, biochemical characteristics, and comorbidities, have not been described in Table 1.
- An analysis of the outcomes related to socio-economic status and pregnancy outcomes is needed.
- "Yuan/year" should be converted to an appropriate international unit.
- It is recommended to include an analysis of a pre-gestational diabetes group.
The regression models in Table 5 should be adjusted after excluding comorbidities and basic biochemical indices to demonstrate that socio-economic status is an independent risk factor for gestational diabetes mellitus.
- A graphical abstract is missing and should be included in the study.

Reviewer 2 ·

Basic reporting

1.Abstract-It gives a different perspective on your findings. Therefore, please highlight your findings clearly and develop the conclusion accordingly.
Introduction-It would be better to provide a strong justification to highlight the importance of this study both nationally and internationally.
Methodology-I think your study design is more likely a descriptive cross-sectional study, as you have collected much of the information from previous notes and the present condition. Also, it is not clear at what point the data is collected: at the completion of the 3rd semester, after childbirth, etc. It would be better to explain this clearly.
I think your study design is more likely a descriptive cross-sectional study, as you have corrected much of the information from previous notes and the present condition. Also, it is not clear at what point the data is collected: at the completion of the 3rd semester, after childbirth, etc. It would be better to explain this clearly.
It is not clear whether you collected data related to physical activities and dietary habits using a standardized questionnaire or by asking a small number of questions.
Results-check your line number 117-119.
What is the justification for using 18.5 kg/m² as the cutoff point to divide the groups? Below 18.5 means they were in the malnourished group. Previous literature has also reported that being over 35 years of age is a risk factor. Additionally, ethnicity is a risk factor for women with GDM. Therefore, it would be better to mention the ethnic background of the women in your study. If you selected only one ethnic group, it is better to mention this as a limitation.
When developing a model, you should include variables that have a linear relationship. Did you check for this? If yes, please present these results as well.
What about history of GDM?Did you include or exclude them
Discussion-It would be better to discuss your findings in a more systematic way and include the limitations, as this study has several
Conclusion-After modifying your results please modify your conclusion

Experimental design

please modify your methodology and the results part as mentioned in above comments

Validity of the findings

There are some concerns regarding the validity of your results. It would be better to seek support from a statistician

Additional comments

Please check your grammar and writing style. It would be better to get support from a professional language editor

---

## Round 0.2 · accepted · Accept

The authors have corrected the key methodological gaps. I therefore recommend the article be accepted for publication.

·

Basic reporting

All elements are acceptable and consistent with the requirements.

Experimental design

All elements are acceptable and consistent with the requirements.

Validity of the findings

All elements are acceptable and consistent with the requirements.

Additional comments

The 'yuan/year' issue still persists

Reviewer 2 ·

Basic reporting

They have addressed comment appropriately.

Experimental design

They have addressed comment appropriately.

Validity of the findings

They have addressed comment appropriately.

Additional comments

They have addressed the comment appropriately. However, it would be better to modify the conclusions in the abstract section.